# Semi-supervised Keypoint Localization

**Olga Moskvyak, Frederic Maire, Feras Dayoub**
School of Electrical Engineering and Robotics
Queensland University of Technology, Australia
`{olga.moskvyak,f.maire,feras.dayoub}@qut.edu.edu`

**Mahsa Baktashmotlagh**
School of Information Technology and Electrical Engineering
The University of Queensland, Australia
`m.baktashmotlagh@uq.edu.au`

## Abstract

Knowledge about the locations of keypoints of an object in an image can assist in fine-grained classification and identification tasks, particularly for the case of objects that exhibit large variations in poses that greatly influence their visual appearance, such as wild animals. However, supervised training of a keypoint detection network requires annotating a large image dataset for each animal species, which is a labor-intensive task. To reduce the need for labeled data, we propose to learn simultaneously keypoint heatmaps and pose invariant keypoint representations in a semi-supervised manner using a small set of labeled images along with a larger set of unlabeled images. Keypoint representations are learnt with a semantic keypoint consistency constraint that forces the keypoint detection network to learn similar features for the same keypoint across the dataset. Pose invariance is achieved by making keypoint representations for the image and its augmented copies closer together in feature space. Our semi-supervised approach significantly outperforms previous methods on several benchmarks for human and animal body landmark localization.

## 1    Introduction

Detecting keypoints helps with fine-grained classification (Guo & Farrell, 2019) and re-identification (Zhu et al., 2020; Sarfraz et al., 2018). In the domain of wild animals (Mathis et al., 2018; Moskvyak et al., 2020; Liu et al., 2019a;b), annotating data is especially challenging due to large pose variations and the need for domain experts to annotate. Moreover, there is less commercial interest in keypoint estimation for animals compared to humans, and little effort is invested in collecting and annotating public datasets.

Unsupervised detection of landmarks[1] (Jakab et al., 2018; Thewlis et al., 2017; 2019) can extract useful features, but are not able to detect perceptible landmarks without supervision. On the other hand, supervised learning has the risk of overfitting if trained only on a limited number of labeled examples. Semi-supervised learning combines a small amount of labeled data with a large amount of unlabeled data during training. It is mostly studied for classification task (van Engelen & Hoos, 2019) but it is also important for keypoint localization problem because annotating multiple keypoints per image is a time-consuming manual work, for which precision is the most important factor. Pseudo-labeling (Lee, 2013) is a common semi-supervised approach where unlabeled examples are assigned labels (called pseudo-labels) predicted by a model trained on a labeled subset. A heuristic unsupervised criterion is adopted to select the pseudo-labeled data for a retraining procedure. More recently, the works of (Dong & Yang, 2019; Radosavovic et al., 2018) apply variations to selection criteria in pseudo-labeling for semi-supervised facial landmark detection. However, there are less variations in facial landmark positions than in human or animal body joints, where there is a high

---

[1]We use terms *keypoints* or *landmarks* interchangeably in our work. These terms are more generic than body joints (used in human pose estimation) because our method is applicable to a variety of categories.

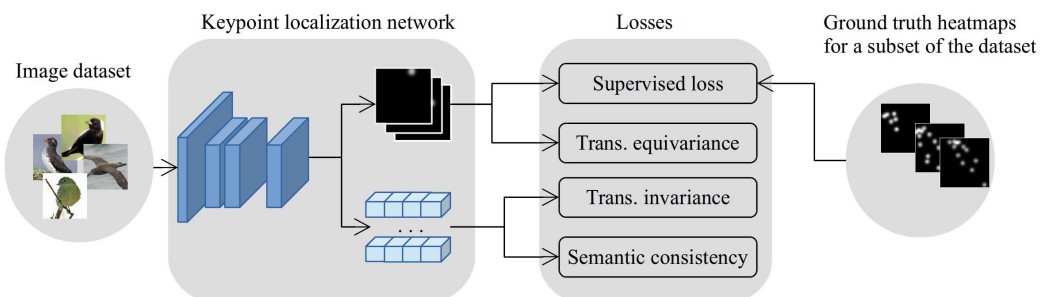

Figure 1: Our semi-supervised keypoint localization system learns a list of heatmaps and a list of semantic keypoint representations for each image. In addition to a supervised loss optimized on the labeled subset of the data, we propose several unsupervised constraints of transformation equivariance, transformation invariance, and semantic consistency.

risk of transferring inaccurate pseudo-labeled examples to the retraining stage that is harmful for the model.

Previous work of (Honari et al., 2018) in semi-supervised landmark detection utilizes additional class attributes and test only on datasets that provide these attribute annotations. Our work focuses on keypoint localization task in a common real-world scenario where annotations are provided for a small subset of data from a large unlabeled dataset. More specifically, we propose a method for semi-supervised keypoint localization that learns a list of heatmaps and a list of semantic keypoint representations for each image (Figure 1). A semantic keypoint representation is a vector of real numbers in a low-dimensional space relative to the image size, and the same keypoints in different images have similar representations. We leverage properties that are specific to the landmark localization problem to design constraints for jointly optimizing both representations.

We extend a transformation consistency constraint of (Honari et al., 2018) to be able to apply it on each representation differently (i.e. transformation equivariant constraint for heatmaps and transformation invariant constraint for semantic representations). Moreover, we formulate a semantic consistency constraint that encourages detecting similar features across images for the same landmark independent of the pose of the object (e.g. an eye in all images should look similar). Learning both representations simultaneously allows us to use the power of both supervised and unsupervised learning.

Our work is motivated by data scarcity in the domain of wild animals, but is not limited to animals, and as well, it is applicable to human body landmarks detection. The contribution of our work is three-fold:

- We propose a technique for semi-supervised keypoint localization that jointly learns keypoint heatmaps and semantic representations optimised with supervised and unsupervised constraints;

- Our method can be easily added to any existing keypoint localization networks with no structural and with minimal computational overhead;

- We evaluate the proposed method on annotated image datasets for both humans and animals. As demonstrated by our results, our method significantly outperforms previously proposed supervised and unsupervised methods on several benchmarks, using only limited labeled data.

The paper is organised as follows. Related work on semi-supervised learning and keypoint localization is reviewed in Section 2. Our proposed method is described in Section 3. Experimental settings, datasets and results are discussed in Section 4.

## 2 RELATED WORK

**Keypoint localization.** Supervised keypoint localization research is driven by a few large datasets with labeled keypoints that span across several common research domains including human pose estimation (Andriluka et al., 2014) and facial keypoints (Sagonas et al., 2016). Challenges in obtaining keypoint annotations have led to the rise in unsupervised landmark localization research. Several unsupervised methods leverage the concept of equivariance which means that landmark coordinates stay consistent after synthetic transformations or in subsequent video frames. Thewlis et al. (2017) propose to learn viewpoint-independent representations that are equivariant to different transformations and Dong et al. (2018) exploit the coherence of optical flow as a source of supervision. Zhang et al. (2018) learn landmark encodings by enforcing constraints that reflect the necessary properties for landmarks such as separability and concentration. Jakab et al. (2018) propose a generative approach where the predicted heatmaps are used to reconstruct the input image from a transformed copy. Recent work (Thewlis et al., 2019) enforce the consistency between instances of the same object by exchanging descriptor vectors. These methods are mostly evaluated on faces of people that have less degrees of freedom during movements and transformations than human or animal body joints. We compare our method to the combination of supervised and aforementioned unsupervised methods in Section 4.

**Semi-supervised learning** is the most studied for the classification task. Pseudo-labeling (Lee, 2013) is a method that uses the model's class predictions as artificial labels for unlabeled examples and then trains the model to predict these labels. Another technique is a consistency regularization which states that realistic perturbations of input examples from unlabeled dataset should not significantly change the output of a neural network. Consistency regularization is used in $\Pi$-model (Laine & Aila, 2017) and further improved by Temporal Ensembling (Laine & Aila, 2017) which maintains an exponential moving average prediction for each training example and Mean Teacher (Tarvainen & Valpola, 2017) that averages model weights instead of model predictions. Recent methods UDA (Xie et al., 2019), ReMixMatch (Berthelot et al., 2020) and FixMatch (Sohn et al., 2020) use a combination of consistency loss, pseudo-labeling and advanced augmentation techniques in addition to color perturbations and spatial transformations. In this work, we investigate adjustments required to apply consistency loss to keypoint localization which we discuss in Section 3.2.

**Semi-supervised learning for keypoint localization.** To the best of our knowledge, there are a few works in semi-supervised keypoint localization. Dong & Yang (2019) build on the pseudo-labeling technique and propose a teacher model and two students to generate more reliable pseudo-labels for unlabeled images. However, the method is evaluated on face landmarks and in cases with high variations of poses, there is a high possibility of inaccurate pseudo-labels that cannot be filtered out and be harmful during the retraining stage. Honari et al. (2018); Ukita & Uematsu (2018) learn keypoints in a semi-supervised manner but utilise extra annotations to guide landmark learning such as action labels (running, jumping) for juman joints or emotion labels (smiling, yawning) for facial keypoint localization. Different from previous work our approach does not use any class labels and learns directly from unlabeled data with high pose variations.

## 3 SEMI-SUPERVISED LEARNING FOR KEYPOINT LOCALIZATION

In this work, we propose a semi-supervised technique for keypoint localization that learns from an image set where ground truth annotations are provided only for a small subset of the dataset. The overall architecture consists of two components: a keypoint localization network (KLN) that outputs keypoint heatmaps of the image, and a keypoint classification network (KCN) that classifies keypoints given a semantic keypoint representation as input. Our method does not pose any constraints on the architecture of the KLN and it can be added to any existing keypoint localization network with minimal modifications.

We optimize heatmaps with the supervised loss and the transformation equivariance constraint. Simultaneously, keypoint representations are optimized with transformation invariance and semantic consistency constraints (Figure 1). We discuss each constraint and related components of the architecture in the next sections.

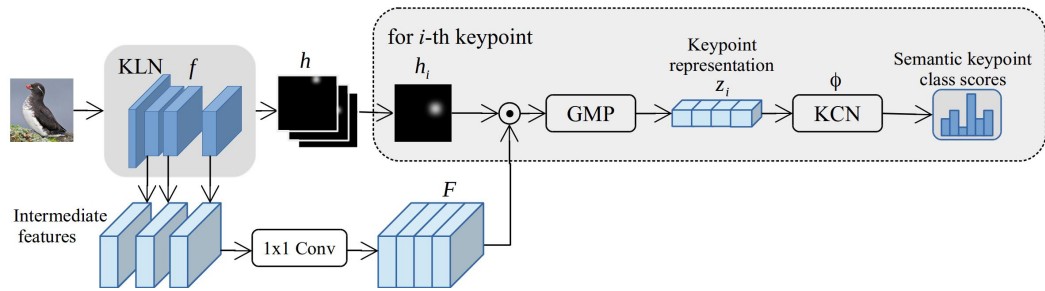

Figure 2: Semantic consistency criteria. Keypoint representation is defined for each keypoint by multiplying a corresponding predicted heatmap $\boldsymbol{h}_i$ with intermediate features $F$. Keypoint representations are classified with a shared network $\phi$ and the feedback is added to the total loss.

### 3.1 SEMANTIC KEYPOINT REPRESENTATIONS

Keypoint heatmaps are optimized to estimate locations of keypoints in the image. However, heatmaps do not carry any information about a semantic type of the keypoint (e.g, a beak or an eye for a bird). In semi-supervised regime, the feedback provided by unlabeled examples are not as effective as the ones coming from labeled examples. To extract useful information from unlabeled images, we propose learning *a semantic keypoint representation*. In particular, keypoint localization network is encouraged to detect similar features for the same semantic keypoint across the dataset by incorporating the feedback from a keypoint representation classifier in the objective function.

Motivation for our approach is that the same keypoints should activate the same feature maps. Let us consider KLN as a function $f(\boldsymbol{x}; \boldsymbol{\theta})$ with an input image $\boldsymbol{x}$ and trainable parameters $\boldsymbol{\theta}$ that outputs heatmaps $\boldsymbol{h} = f(\boldsymbol{x}; \boldsymbol{\theta})$. We collect intermediate feature maps from KLN, upscale them to the spatial dimension of output heatmaps, concatenate by channels, and pass through a convolutional layer with $C$ filters of size one (Figure 2). The resulting feature map $F$ has the shape $(C, H, W)$. Then, feature maps $F$ are element-wise multiplied with each keypoint heatmap $\boldsymbol{h}_i, i \in \{1, ..., K\}$ seperately to mask out activations corresponding to the detected keypoint. The output of this operation is $K$ feature maps of size $(C, H, W)$. Global Max Pooling (GMP) is applied over feature maps to keep the highest value for each channel. We call the produced vector $\boldsymbol{z}_i = \text{GMP}(F \odot \boldsymbol{h}_i)$ for each keypoint $i \in \{1, ..., K\}$ a semantic keypoint representation.

Finally, we pass keypoint representations to a simple KCN ($\phi$) which is a fully connected network with an input and an output layer for classification with cross-entropy loss. The feedback from the cross-entropy loss makes up a *semantic consistency (SC)* loss:

$$\mathcal{L}_{\text{sc}}(\boldsymbol{x}) = -\frac{1}{K} \sum_{i=1}^{K} \hat{y}_i \log(\phi(\boldsymbol{z}_i)) \qquad (1)$$

where $\hat{y}$ is a vector of ground truth semantic labels for keypoints because the order of keypoints in a heatmap is fixed.

One advantage of our method is its efficiency as it only adds a small number of parameters to the network to address the task of keypoint representation classification. Specifically, KCN is a small fully connected network shared between keypoints and it has less than a thousand of parameters depending on the number of keypoints. Our approach is related to attention modules (Vaswani et al., 2017; Hu et al., 2020) as our network has the ability to focus on a subset of features using element-wise multiplication with heatmaps. However, our model uses this attention-based mechanism to learn additional keypoint representations from unlabeled data by optimizing a set of unsupervised losses.

### 3.2 TRANSFORMATION CONSISTENCY CONSTRAINT

The difference between keypoint heatmaps and semantic keypoint representations is that the former is transformation equivariant and the latter is transformation invariant. In other words, the output

heatmaps should be consistent with viewpoint variations of the image while keypoint representations should be preserved for all different transformations of the image. We call this property a *transformation consistency constraint.*

**Transformation equivariance (TE)** enforces a commutative property on the landmark localization and augmentation operations that include spatial transformation (e.g, rotations and translations), meaning that the order of applying these two operations does not matter. Let $g(\cdot, \boldsymbol{s})$ be an augmentation function with augmentation parameters $\boldsymbol{s}$ which are not trainable and sampled randomly each time. Transformation equivariance constraint is formulated as: $f \circ g(\boldsymbol{x}) = g \circ f(\boldsymbol{x})$. We measure *a transformation equivariance loss* $\mathcal{L}_{\text{te}}$ over predicted heatmaps by squared Euclidean distance:

$$\mathcal{L}_{\text{te}}(\boldsymbol{x}; \boldsymbol{\theta}) = \mathbb{E}_{\boldsymbol{x}}\Big[||f(g(\boldsymbol{x}, \boldsymbol{s}); \boldsymbol{\theta}) - g(f(\boldsymbol{x}; \boldsymbol{\theta}), \boldsymbol{s})||^2\Big] \tag{2}$$

Note that, after applying a transformation, some landmarks may go outside of the image boundary, and cause the visibility issue. This problem is alleviated in our formulation by applying the same transformation to an image. This is different from equivariant landmark transformation (ELT) loss proposed by Honari et al. (2018) which computes an inverse transformation instead. In essence, inverse transformation cannot bring these landmarks back meaning that inverse transformation does not output the original image. Our approach avoids this issue.

**Transformation invariance (TI)** of keypoint representations is enforced by pulling corresponding vectors for the image and its augmented copy closer together. First, we concatenate keypoint representations in one vector to get a holistic representation $\boldsymbol{z}$ of the image $\boldsymbol{x}$:

$$\boldsymbol{z} = [\boldsymbol{z}_1, \boldsymbol{z}_2, ..., \boldsymbol{z}_K]. \tag{3}$$

We apply a random spatial transformation to the input image to get image $\boldsymbol{x}'$, compute keypoint representations $\boldsymbol{z}'_1, \boldsymbol{z}'_2, ..., \boldsymbol{z}'_K$, and concatenate them to get a vector $\boldsymbol{z}'$. Finally, we enforce pose invariance by penalizing a distance between representations of original and transformed images and formulate *a transformation invariance loss* $\mathcal{L}_{\text{ti}}$:

$$\mathcal{L}_{\text{ti}}(\boldsymbol{x}, \boldsymbol{x}') = \mathbb{E}_{\boldsymbol{x}, \boldsymbol{x}'}\Big[||\boldsymbol{z} - \boldsymbol{z}'||^2\Big] \tag{4}$$

The overall objective is the weighted sum of losses:

$$\mathcal{L} = \lambda_1 \mathcal{L}_{\text{sup}} + \lambda_2 \mathcal{L}_{\text{sc}} + \lambda_3 \mathcal{L}_{\text{te}} + \lambda_4 \mathcal{L}_{\text{ti}} \tag{5}$$

where $\mathcal{L}_{\text{sup}}$ is a supervised mean squared error between predicted and ground truth heatmaps for the labeled subset. Parameters $\lambda_i$ are defined experimentally.

## 4 EXPERIMENTS

### 4.1 DATASETS

We evaluate our method on two datasets with annotated human body joints and two datasets of wild animals.

**MPII** Human Pose dataset (Andriluka et al., 2014) is a collection of images showing people doing real-world activities with annotations for the full body. Due to the fact that test annotations are not released publicly, we use training and validation splits of MPII in our experiments. We use 10,000 images for training to speed up experiments as we run multiple training runs for each subset of labeled examples. Our validation and test sets consist of 3,311 and 2,958 images respectively. Annotations contain coordinates for 16 body joints with a visibility flag.

**LSP** (Leeds Sports Pose) (Johnson & Everingham, 2010; 2011) dataset is a collection of annotated images with people doing sports such as athletics, badminton or soccer. Each image has been annotated with 14 joint locations. We use 10,000 images from extended (Johnson & Everingham, 2011) version for training and 2,000 images from original (Johnson & Everingham, 2010) dataset for testing and validation.

**CUB-200-2011** (Welinder et al., 2010) is a dataset of 200 fine-grained classes of bird species. We split dataset into training, validation and testing with disjoint classes so test classes does not appear

during training. First 100 classes are used for training (5,864 images), 50 classes for validation (2,958 images) and the last 50 classes (2,966 images) for testing. Each image is annotated with 15 body keypoints such as beak, left eye and throat. We use class label only for splitting the dataset and do not use it anywhere in out method.

**ATRW** (Li et al., 2019) is a dataset of Amur tigers images captured in multiple wild zoos in unconstrained settings. Professionals annotated 15 skeleton keypoints for each tiger. We use 3,610 images for training, 516 for validation and 1,033 for testing with annotations provided by authors. This dataset is more challenging than birds as four-legged animals exhibit more pose variations.

Training set for each dataset is split into labeled and unlabeled subsets by randomly picking 5%, 10%, 20% or 50% of the training examples and discarding the labels for the rest of the data. The procedure is repeated three times so all experiments are run three times to obtain the mean and standard deviation of the results. Validation and test sets are fixed for all experiments. Validation set is used to tune hyperparameters and test set is used to report the final results. The order of the keypoints is explicitly defined in annotations and is fixed for the training and inference.

The evaluation metric is PCK (probability of correct keypoint) from (Yang & Ramanan, 2013) where a keypoint is considered correctly localized if it falls within $\alpha l$ pixels of the ground truth position ($\alpha$ is a constant and $l$ is the maximum side of the bounding box). The PCK@0.1 ($\alpha = 0.1$) score is reported for LSP, CUB-200-2011 and ATRW datasets. For MPII we use an adaptation (Andriluka et al., 2014) which is PCKh (head-normalized probability of correct keypoint) where $l$ is the head size that corresponds to 60% of the diagonal length of the ground truth head bounding box (provided in the MPII annotations).

## 4.2 IMPLEMENTATION DETAILS

Images are resized to the input size $256 \times 256$ and heatmaps are predicted at size $64 \times 64$. We adapt HRNet-32 (Sun et al., 2019) architecture as KLN because it is originally designed for keypoint localization and retains features at high spatial dimension (e.g. $64 \times 64$ for the input of size $256 \times 256$). We collect intermediate features at the output of each multi-scale subnetwork, after concatenation we get 352 channels and then apply 64 convolutional filters of size one. GMP results in representations of length 64 for each keypoint. We also experimented with collecting more features from different layers but it did not improve the performance. KCN is a fully connected network that accepts keypoint representation of size 64 and classifies keypoints based on their semantic labels (from 10 to 17 depending on the dataset).

We use perspective transformations as an augmentation function $g$ where parameters $s$ of the transformation are sampled randomly using a method from (Moskvyak & Maire, 2017) to avoid extreme warping. We also experimented with simple affine transformations but perspective gave better results most likely due to higher variability of transformations.

Unsupervised losses may hurt the learning at the beginning because output heatmaps and intermediate feature maps are random during first epochs. A possible solution is to vary the contribution of unsupervised losses according to a predefined strategy. To avoid tuning many hyperparameters, our semi-supervised approach uses ground truth heatmaps in unsupervised losses for the labeled samples in a batch. This approach has only one hyperparameter - percentage of the labeled samples in a batch. We found that there is enough feedback from labeled examples when the batch has 50% of labeled and 50% of unlabeled examples.

We adopt Adam (Kingma & Ba, 2015) optimizer with learning rate $10^{-4}$ for all experiments. Models are trained until the accuracy on the validation set has stopped improving. The weights of loss components were determined experimentally $(\lambda_1, \lambda_2, \lambda_3, \lambda_4) = (10^3, 0.5, 10^2, 10^2)$. We provide the sensitivity analysis in Section 4.

## 4.3 RESULTS

**Comparison with the supervised baseline.** We train HRNet-32 (Sun et al., 2019) with the supervised loss as a baseline from the official implementation on the labeled subsets with 5%, 10%, 20%, 50% and 100% of the dataset. The baseline performance decreases significantly when the amount of training data is reduced on human poses and tigers datasets (Table 1). On birds dataset, we observe

Table 1: PCK@0.1 score for keypoint localization with different percentage of labeled images. We report mean and standard deviation from three runs for different randomly sampled labeled subsets. Pseudo-labeled (PL) baseline is not evaluated for 100% of labeled data because there is no unlabeled data to generate pseudo-labels for.

| Method | Percentage of labeled images | | | | |
| --- | --- | --- | --- | --- | --- |
| | 5% | 10% | 20% | 50% | 100% |
| **Dataset 1: MPII** | | | | | |
| HRNet (Sun et al., 2019) | 66.22±1.60 | 69.18±1.03 | 71.83±0.87 | 75.73±0.35 | 81.11±0.15 |
| PL (Radosavovic et al., 2018) | 62.44±1.75 | 64.78±1.44 | 69.35±1.11 | 77.43±0.48 | - |
| ELT (Honari et al., 2018) | 68.27±0.64 | 71.03±0.46 | 72.37±0.58 | 77.75±0.31 | 81.01±0.15 |
| Gen (Jakab et al., 2018) | 71.59±1.12 | 72.63±0.62 | 74.95±0.32 | 79.86±0.19 | 80.92±0.32 |
| **Ours** | **74.15±0.83** | **76.56±0.48** | **78.46±0.36** | **80.75±0.32** | **82.12±0.14** |
| **Dataset 2: LSP** | | | | | |
| HRNet (Sun et al., 2019) | 40.19±1.46 | 45.17±1.15 | 55.22±1.41 | 62.61±1.25 | 72.12±0.30 |
| PL (Radosavovic et al., 2018) | 37.36±1.89 | 42.05±1.68 | 48.86±1.23 | 64.45±0.96 | - |
| ELT (Honari et al., 2018) | 41.77±1.56 | 47.22±0.91 | 57.34±0.94 | 66.81±0.62 | 72.22±0.13 |
| Gen (Jakab et al., 2018) | 61.01±1.41 | 67.75±1.00 | 68.80±0.91 | 69.70±0.77 | 72.25±0.55 |
| **Ours** | **66.98±0.94** | **69.56±0.66** | **71.85±0.33** | **72.59±0.56** | **74.29±0.21** |
| **Dataset 3: CUB-200-2011** | | | | | |
| HRNet (Sun et al., 2019) | 85.77±0.38 | 88.62±0.14 | 90.18±0.22 | 92.60±0.28 | 93.62±0.13 |
| PL (Radosavovic et al., 2018) | 86.31±0.45 | 89.51±0.32 | 90.88±0.28 | 92.78±0.27 | - |
| ELT (Honari et al., 2018) | 86.54±0.34 | 89.48±0.25 | 90.86±0.13 | 92.26±0.06 | 93.77±0.18 |
| Gen (Jakab et al., 2018) | 88.37±0.40 | 90.38±0.22 | 91.31±0.21 | 92.79±0.14 | 93.62±0.25 |
| **Ours** | **91.11±0.33** | **91.47±0.36** | **92.36±0.30** | **92.80±0.24** | **93.81±0.13** |
| **Dataset 4: ATRW** | | | | | |
| HRNet (Sun et al., 2019) | 69.22±0.87 | 77.55±0.84 | 86.41±0.45 | 92.17±0.18 | 94.44±0.10 |
| PL (Radosavovic et al., 2018) | 67.97±1.07 | 75.26±0.74 | 84.69±0.57 | 92.15±0.24 | - |
| ELT (Honari et al., 2018) | 74.53±1.24 | 80.35±0.96 | 87.98±0.47 | 92.80±0.21 | 94.75±0.14 |
| Gen (Jakab et al., 2018) | 89.54±0.57 | 90.48±0.49 | 91.16±0.13 | 92.27±0.24 | 94.80±0.13 |
| **Ours** | **92.57±0.64** | **94.29±0.66** | **94.49±0.36** | **94.63±0.18** | **95.31±0.12** |

only a small decrease in the baseline score (Table 1). We explain it by the fact that there are more variations in poses of four-legged animals and human body joints than of birds. Supervised results on MPII are lower than the official ones because the training set is smaller and we do not include additional tricks during training (e.g. half body transforms) and testing (post-processing and averaging over flipped images).

Our method significantly improves the baseline on all datasets (Table 1). Our proposed unsupervised constraints are the most beneficial for low data regimes with 5%, 10% and 20% labeled images. For example, our method increases the score from 40% to 66% on LSP dataset with 5% of labeled data. On the challenging tigers dataset, our approach reaches the score of 92% trained with only 5% labeled examples when the supervised model shows the score 69% while trained on the same labeled data. Experiments show that the influence of additional unsupervised losses decreases when more labeled examples are added to the training. Experiments show that our method on 100% labeled data outperforms the supervised baseline by a small margin because by learning supplementary semantic keypoint representations with unsupervised losses the model learns to generalize better.

**Comparison with the pseudo-labeled baseline.** We apply pseudo-labeled (PL) method from Radosavovic et al. (2018) on our datasets (Table 1). We use the same model HRNet-32 as in all our experiments for a fair comparison. Overall, the pseudo-labeled baseline is inferior to our method on all datasets used in our study. We explain it by the fact that Radosavovic et al. (2018) trained on datasets that are by order of magnitude larger than our data so models pretrained on the labeled subset are already good enough to generate reliable pseudo-labels.

Table 2: Ablation study on LSP. We isolate gains from each unsupervised loss to evaluate the contribution of each constraint. SC - semantic consistency TE - transformation equivariance and TI - transformation invariance. TE is not evaluated separately because it does not optimize both representations. Results are reported on one run.

| Unsupervised losses | Percentage of labeled images | | | | |
|---|---|---|---|---|---|
| | 5% | 10% | 20% | 50% | 100% |
| TE + TI + SC | 66.32 | 69.09 | 71.62 | 72.19 | 74.44 |
| TE + TI | 46.76 | 55.18 | 64.01 | 67.54 | 72.11 |
| SC | 64.74 | 67.43 | 69.65 | 70.61 | 72.85 |
| TI + SC | 65.23 | 68.11 | 70.12 | 71.28 | 73.56 |
| TE + SC | 65.78 | 68.51 | 70.56 | 71.77 | 73.89 |
| TI | 43.62 | 53.74 | 61.12 | 65.32 | 71.80 |
| Supervised baseline | 39.16 | 44.36 | 54.23 | 61.73 | 71.91 |

Table 3: Influence of the amount of unlabeled data. We train our model with the fixed 5% of labeled data and vary the amount of unlabeled data. Results are reported on one run. The results are compared with the supervised baseline (SB) trained with the same labeled data.

| Dataset | Percentage of unlabeled images | | | | SB |
|---|---|---|---|---|---|
| | 10% | 20% | 50% | 100% | |
| CUB-200-2011 | 87.01 | 88.33 | 89.44 | 91.34 | 85.33 |
| ATRW | 72.04 | 76.65 | 86.56 | 93.02 | 69.84 |

**Comparison with related methods.** We compare our approach with previously proposed semi-supervised and unsupervised methods for landmark detection (Table 1). The equivariant landmark transformation (ELT) loss from (Honari et al., 2018) forces a model to predict equivariant landmarks with respect to transformations applied to an image. ELT loss gives a small improvement over the baseline model and is inferior to our method on all datasets. Jakab et al. (2018) learn keypoints without supervision by encouraging the keypoints to capture the geometry of the object by learning to generate the input image given its predicted keypoints and an augmented copy. For a fair comparison we inject the models from Jakab et al. (2018) into our training pipeline and add the supervised loss for the labeled examples in each batch. All other parameters are kept the same including augmentation, subsets of data and training schedule. We observe that the generation approach improves over ELT loss and the baseline however it is inferior to our method. The generation approach also introduces more parameters (in the reconstruction part of the network) than our approach that adds only a small keypoint classifier network.

**Ablation study.** We investigate the influence of different loss components of our methods on LSP dataset (Table 2). At first, we remove semantic consistency loss component (Eq. 1) and observe the significant drop in the score especially in low labeled data regime. For example, with 5% of labeled data the score drops from 66% when trained with the combination TE + TI + SC to 46% for the combination TE + TI. When we return semantic consistency and remove transformation consistency losses (Eq. 2, 4), the results are reduced slightly. The results of ablation study shows that the semantic consistency loss component is more influential than the transformation consistency. Both TE and TI losses contribute to the performance gain and their combination achieves better results than each loss separately. We argue that our TE loss is an improvement over ELT loss (Honari et al., 2018). We replaced our TE loss with an inverse transformation loss of Honari et al. (2018) in our framework, and applied it on ATRW and CUB-200-2011 datasets with 20% of labeled data. We observed that the score decreased by 1% on both datasets.

We also analyse the influence of the amount of unlabeled data in our method (Table 3). We conduct experiments where the amount of labeled examples is fixed at 5% and the number of unlabeled examples is reduced to 50%, 20% and 10% of the number of original unlabeled samples. We observe that the score goes down as the amount of unlabeled data is reduced. Our method outperforms the supervised score only by a small margin with 10% of unlabeled data. We conclude that the number of unlabeled examples plays an important role in training with our unsupervised losses.

We conduct an ablation study to get an insight on using ground truth heatmaps in unsupervised losses. Experiments on the LSP dataset show a decrease of 1-2% in the score for all cases when ground truth heatmaps are not used (Table 4). The results prove the benefit of using the signal from available ground truth heatmaps.

Table 4: An ablation study of using ground truth heatmaps in unsupervised losses on LSP dataset. Results are reported on one run.

| Method | Percentage of labeled images | | | | |
| --- | --- | --- | --- | --- | --- |
| | 5% | 10% | 20% | 50% | 100% |
| With g/t heatmaps | 66.32 | 69.09 | 71.62 | 72.19 | 74.44 |
| Without g/t heatmaps | 64.75 | 67.27 | 69.91 | 70.55 | 73.65 |

**Sensitivity analysis of weight loss components.** We fixed the weight $\lambda_1 = 10^3$ and tested weights: $\lambda_2 = (0.1, 0.5, 1.)$, $\lambda_3 = (10^1, 10^2, 10^3)$ and $\lambda_4 = (10^1, 10^2, 10^3)$. The ranges of weight values are different due to differences in scales for mean squared error and cross-entropy loss. Experiments on LSP dataset show that our method is not sensitive to variations of TE ($\lambda_3$) and TI ($\lambda_4$) losses (Figure 3). The most notable drop in accuracy is observed when the weight of SC loss ($\lambda_2$) is reduced to 0.1 and the accuracy is at the same level when $\lambda_2$ equals 0.5 and 1.0. We select the combination of $(\lambda_2, \lambda_3, \lambda_4) = (0.5, 10^2, 10^2)$ that achieves the highest score.

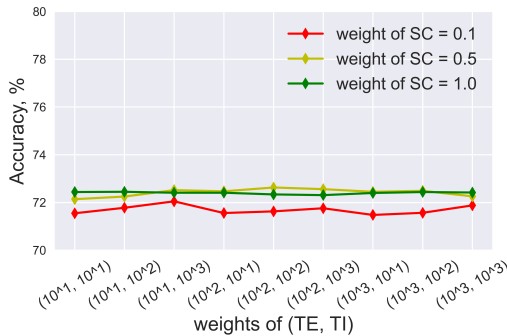

Figure 3: Sensitivity analysis of the weights of loss components.

**Analysis of keypoint representations.** We analyze the learned keypoint representation with t-SNE (van der Maaten & Hinton, 2008). The t-SNE algorithm maps a high dimensional space (64 dimensions in our case) into a two-dimensional while preserving the similarity between points. The t-SNE plot for the keypoint representations of LSP test set (Figure 4) shows that representations for the same keypoints are clustered together.

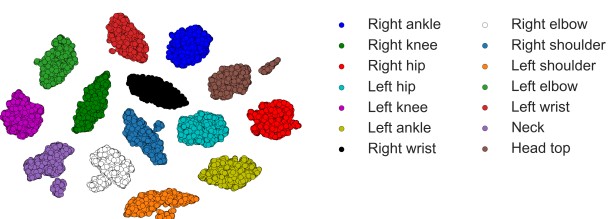

Figure 4: tSNE visualization of keypoint embeddings for human body landmarks on LSP test set.

## 5 CONCLUSION

We presented a new method for semi-supervised keypoint localization. We show that reliable keypoints can be obtained with a limited number of labeled examples. This is achieved by learning semantic keypoint representations simultaneously with keypoint heatmaps using a set of unsupervised constraints tailored for the keypoint localization task. We applied our method to predict human body joints and animal body keypoints and demonstrated that it outperforms current supervised and unsupervised methods. Moreover, it reaches the same performance as the model trained on the whole labeled dataset with only 10% of labeled images on tigers ATRW dataset and with 50% labeled images on challenging human poses LSP dataset. We plan to investigate the applicability of our method to domain adaptation for keypoint localization in the future work.

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
