# OpenReview forum: "Semi-supervised Keypoint Localization"
_ICLR.cc/2021/Conference — ICLR 2021 Poster_

### Official Review · AnonReviewer3 · 2020-10-26
**This paper proposes a model for landmark/keypoint localization trained in a semi-supervised way**

**Rating:** 6
**Confidence:** 3

**Review:**

It can be applied to point heatmaps based network by adding a semantic representation learn by a three loss terms: one supervised and two semi-supervised. The proposed architecture combines a Keypoint Localization Network with a Keypoint Classification Network. Experiments are achieved on four public datasets.

The main contribution of the paper is the model and losses proposed to train, in a semi-supervised way, the network.

Contributions are clearly stated and validated.
The idea of using intermediate features to produce a map that will select keypoint features with an element-wise product with the heatmap is good. Its looks like a kind of attention module. Additional information should be provide to explain the differences: this is mandatory
The transformation consistency constraints are also good ideas and the modified transformation equivariance is a smart trick.
Experiments have been achieved in order to compare the proposed semi-supervised model with other semi-supervised models (2 are selected). Results are good.
Moreover, an ablation study is proposed. It reports the effect of each unsupervised loss. It should be interesting to add the score of the model without unsupervised loss. It will, for example show if for 100% sample trained, adding unsupervised loss improves the method. A visualisation ok keypoint embedding using the tSNE technic is also proposed. I'am not sure that this figure gives usefull information for the study if not compared to the one obtained by only supervised training.

The implementation details points that "unsupervised loss may hurt the learning at the beginning" and proposes to use ground truth heatmaps. This is not really an implementation detail to my point of view are more information and experiments about this trick should be given.

---

> ### Author Response · Authors · 2020-11-20
> **Additional ablation studies and response to questions**
>
> We thank the reviewer for the positive comments and constructive feedback. We have completed the additional ablation study and addressed the questions below. The paper is revised accordingly.
>
> In response to the question of comparing our method to attention models: Similar to attention modules (Vaswani et al., 2017, Ramachandran et al., 2019, Hu et al., 2020), our network has the ability to focus on a subset of features using element-wise multiplication with the heatmaps. However, our model uses the attention-based mechanism to learn additional keypoint representations (apart from the main heatmap output) from unlabeled data by optimizing a set of unsupervised losses.
>
> In response to the reviewer’s comment on providing the results of our framework without the unsupervised losses, we added a supervised baseline to Table 2 for convenient comparison. The results in Table 2 show clearly that adding unsupervised losses improves the supervised score.
> Table 2 (shown below) is updated in the revised paper accordingly.
>
> **Updated Table 2**
>
> | Unsupervised losses 	| 5% 	| 10% 	| 20% 	| 50% 	| 100% 	|
> |-	|:-:	|:-:	|:-:	|:-:	|:-:	|
> | TE + TI + SC 	| 66.32 	| 69.09 	| 71.62 	| 72.19 	| 74.44 	|
> | TE + TI 	| 46.76 	| 55.18 	| 64.01 	| 67.54 	| 72.11 	|
> | SC 	| 64.74 	| 67.43 	| 69.65 	| 70.61 	| 72.85 	|
> | TI + SC 	| 65.23 	| 68.11 	| 70.12 	| 71.28 	| 73.56 	|
> | TE + SC 	| 65.78 	| 68.51 	| 70.56 	| 71.77 	| 73.89 	|
> | TI 	| 43.62 	| 53.74 	| 61.12 	| 65.32 	| 71.80 	|
> | Supervised baseline 	| 39.16 	| 44.36 	| 54.23 	| 61.73 	| 71.91 	|
>
> We conduct an ablation study to provide an insight on using available ground truth heatmaps in unsupervised losses. Experiments on the LSP dataset show a decrease of 1-2% in the score for all cases when ground truth heatmaps are not used. The results prove the benefit of using the signal from available ground truth heatmaps.
> The following results are added as Table 4 to the paper:
>
> | Method 	| 5% 	| 10% 	| 20% 	| 50% 	| 100% 	|
> |-	|:-:	|:-:	|:-:	|:-:	|:-:	|
> | With g/t heatmaps 	| 66.32 	| 69.09 	| 71.62 	| 72.19 	| 74.44 	|
> | Without g/t heatmaps 	| 64.75 	| 67.27 	| 69.91 	| 70.55 	| 73.65 	|
>
>
>
> References:
>
> Hu, J., Shen, L., Albanie, S., Sun, G., & Wu, E. (2020). Squeeze-and-Excitation Networks. IEEE Transactions on Pattern Analysis and Machine Intelligence, 42, 2011-2023.
>
> Ramachandran, P., Parmar, N., Vaswani, A., Bello, I., Levskaya, A., & Shlens, J. (2019). Stand-Alone Self-Attention in Vision Models. NeurIPS.
>
> Vaswani, A., Shazeer, N., Parmar, N., Uszkoreit, J., Jones, L., Gomez, A.N., Kaiser, L., & Polosukhin, I. (2017). Attention is All you Need. NIPS.

---

### Official Review · AnonReviewer1 · 2020-10-28
**Official Blind Review1**

**Rating:** 7
**Confidence:** 4

**Review:**

Summary:

This paper presents an interesting method for semi-supervised keypoint localization, that jointly learns the keypoint heatmaps and pose-invariant keypoint representations. The model is trained semi-supervised by applying transformation consistency and semantic consistency constraints. The proposed method is evaluated on several benchmarks and it significantly outperforms other semi- & un- supervised methods.

##################################################################

Reasons for score:

The paper is mostly clear and well-motivated. The authors develop a novel approach to tackle the problem of semi-supervised keypoint localization. The reviewer appreciates the novelty and ingenuity of this approach. It achieves the state-of-the-art performance, compared to other semi-supervised methods. The proposed method is easy-to-implement and can be added to any existing keypoint localization networks. Also the extra keypoint classification branch is only used for training and can be discarded during inference.

There is still room for greatly improving the experiment section. Hopefully the authors can address the concerns in the rebuttal period. One major problem is the missing comparisons with pseudo-labeling baselines (Dong & Yang, 2019; Radosavovic et al., 2018). Another limitation of this work is the relatively narrow scope. The paper only focuses on one application of semi-supervised keypoint localization. However, I believe such techniques may also applicable to other tasks.

##################################################################

Pros:

1.The proposed method is very interesting and novel. The authors propose to add a keypoint classification branch, and design several well-motivated consistency losses.
2.The paper is clear and well-written.
3.The proposed method is easily added to any existing keypoint localization networks, which means many previous works can be jointly trained in a much larger unlabeled dataset and it may enhance the robustness and performance of the models.

##################################################################

Cons:

1. One major problem is the missing comparisons with pseudo-labeling baselines (Dong & Yang, 2019; Radosavovic et al., 2018). The reviewer believes that the comparisons are critical in showing the effectiveness of the proposed method.
2. The authors said “where there is a high risk of transferring inaccurate pseudo-labeled examples to the retraining stage that is harmful for the model.”However, no evidence is provided in the experiment sections.
3. Although the proposed method provides several ablation studies, the reviewer suggests to consider adding the following experiments to enhance the quality of the paper:

(1)	When percentage of labeled images is 100%, which is totally supervised, the proposed method already outperforms the baseline. Is this method also applicable to improving the performance of fully-supervised cases? Please analyze why.

(2) Ablation study of only adding the keypoint classification branch (without the consistency losses).

(3) Lacking of ablation study about TC in Table 2. The effectivess of transformation equivariance and transformation invariance should be considered separately.


##################################################################

Questions during rebuttal period:

1.In Table3, it reads“Our method outperforms the supervised score only by a small margin with 10% of unlabelled data.” It would be more convenient, if the supervised scores are also listed in the Table.

Some typos:

(1) 4.1 ATRW: where l is is the head → where l is the head
(2) Table 2： Usupervised losses → Unsupervised losses

---

> ### Author Response · Authors · 2020-11-20
> **Results of pseudo-labeled baseline and updated ablation study**
>
> We thank the reviewer for the constructive comments. We appreciate the suggestion of additional baseline and supplementary ablation studies. We conducted the proposed experiments, and the manuscript is revised accordingly.
>
>  1. We obtained the results of pseudo-labeled baseline (Radosavovic et al., 2018) on our datasets and added the results to Table 1. Overall, our approach outperforms the pseudo-labeled baseline on all datasets used in our study. One possible explanation is that  Radosavovic et al. (2018) used datasets that are by order of magnitude larger than our data. For example,  COCO (Lin et al., 2014) consists of 80,000 labeled and 120,000 unlabeled images vs 10,000 total images in our LSP dataset. Therefore models pretrained on the labeled subset are already good enough to generate reliable pseudo-labels.
>
> 2. Results of the pseudo-labeled baseline from the previous question provide the evidence for our claim: “where there is a high risk of transferring inaccurate pseudo-labeled examples to the retraining stage that is harmful for the model.”
> For example, the supervised baseline only achieves the score of 40% on LSP with 5% of labeled data. The score of pseudo-labeled baseline drops to 37% due to low-quality pseudo-labels from the supervised model.
>
> 3. We conducted the suggested ablation studies to enhance the quality of the paper:
>
> (1) We observe in the experiments that our method on 100% labeled data outperforms the supervised baseline by a small margin. This can be due to the fact that by learning supplementary semantic keypoint representations with unsupervised losses, the model learns to generalize better.
>
> (2-3) We analyzed the influence of each unsupervised loss separately. As opposed to the original submission where the transformation equivariance (TE) and invariance (TI) are combined as TC (transformation consistency), in the updated ablation study, we treat them separately and evaluate the contribution of each constraint on the performance individually. We evaluate all possible combinations except single TE separately because it does not optimize both representations. The results of ablation study are added to Table 2.
>
> **Updated Table 2**
>
> | Unsupervised losses 	| 5% 	| 10% 	| 20% 	| 50% 	| 100% 	|
> |-	|:-:	|:-:	|:-:	|:-:	|:-:	|
> | TE + TI + SC 	| 66.32 	| 69.09 	| 71.62 	| 72.19 	| 74.44 	|
> | TE + TI 	| 46.76 	| 55.18 	| 64.01 	| 67.54 	| 72.11 	|
> | SC 	| 64.74 	| 67.43 	| 69.65 	| 70.61 	| 72.85 	|
> | TI + SC 	| 65.23 	| 68.11 	| 70.12 	| 71.28 	| 73.56 	|
> | TE + SC 	| 65.78 	| 68.51 	| 70.56 	| 71.77 	| 73.89 	|
> | TI 	| 43.62 	| 53.74 	| 61.12 	| 65.32 	| 71.80 	|
> | Supervised baseline 	| 39.16 	| 44.36 	| 54.23 	| 61.73 	| 71.91 	|
>
>
> In response to the question, we have updated Table 3 in the revised paper with the supervised baseline (SB) for convenient comparison:
>
> **Updated Table 3**
>
> | Dataset 	| 10% 	| 20% 	| 50% 	| 100% 	| SB 	|
> |-	|:-:	|:-:	|:-:	|:-:	|:-:	|
> | CUB-200-2011 	| 87.01 	| 88.33 	| 89.44 	| 91.34 	| 85.33 	|
> | ATRW 	| 72.04 	| 76.65 	| 86.56 	| 93.02 	| 69.84 	|
>
>
> We fixed the typos in the revised version.
> We thank the reviewer for the suggestion to expand the scope and apply our method to other tasks. We will consider applying it to other vision tasks in future work.
>
> References:
>
> Tsung-Yi Lin, M. Maire, Serge J. Belongie, J. Hays, P. Perona, D. Ramanan, P. Dollar, and C. L.Zitnick. Microsoft coco: Common objects in context. InProc. ECCV, 2014.
>
> lija Radosavovic, P. Dollar, Ross B. Girshick, Georgia Gkioxari, and Kaiming He. Data distillation: Towards omni-supervised learning. In Proc. CVPR, 2018

---

> > ### Comment · AnonReviewer1 · 2020-11-23
> > **Thanks for clarification**
> >
> > My questions were well addressed. I think this work is solid and I recommend acceptance.

---

### Official Review · AnonReviewer4 · 2020-10-28
**SEMI-SUPERVISED KEYPOINT LOCALIZATION**

**Rating:** 6
**Confidence:** 4

**Review:**

**Summary**

The paper presents an approach to keypoint localization (to retrieve people/animals pose) combining labeled and unlabeled data. Features are extracted and concatenated into a single descriptor per keypoints, by multiplying feature maps and heatmaps and max-pooling over the spatial domain, and used for semantic classification. Images are transformed with simple perspective augmentations. The non-supervised part comes in enforcing that keypoint representations for unlabeled images remain close.

**Pros**

* Very simple formulation.
* Thorough evaluation on four datasets with strong results.
* Informative ablation tests.

**Cons**

* Relatively light on contributions, as the paper is quite simple.

**Details**

Please cite Adam (mentioned but not cited).

The baselines/datasets seem sufficient, but I might have missed relevant works (I do not work in this field).

"Dialated": dilated

---

P.S. I received an automated email complaining that my review was too short. The paper is nice, straightforward, and easy to follow. The evaluation seems thorough. I did not find any mistakes. I don't have too much to say about it. My review is positive and, I hope, constructive.

---

> ### Author Response · Authors · 2020-11-20
> **We thank the reviewer for the positive feedback**
>
> We thank the reviewer for the positive feedback, and suggestions that helped improve our paper.
> We took into account all the minor comments of the reviewer in the revised paper.

---

### Official Review · AnonReviewer2 · 2020-10-31
**Review for "semi-supervised keypoint localization"**

**Rating:** 5
**Confidence:** 4

**Review:**

** Paper Summary **

This paper presents semi-supervised keypoint localization networks and loss functions to overcome the need for the labeled keypoint data for that task. It simultaneously generates keypoint heatmaps and pose invariant keypoint representations, where these representations were separately used to enforce translation equivariance, and translation invariance, and semantic consistency, respectively. The proposed method attains the improvement on several benchmarks for human and animal body landmark localization.

** Paper Strength **

+ Learning keypoints with a minimal supervision is an essential step for numerous applications, thus solving such a problem is important.
+ Using two kinds of representations, i.e., heatmap and keypoint feature, makes sense, enabling to apply different loss functions that achieve different aspects, e.g., translation equivariance and translation invariance, and semantic consistency.
+ Compared to other previous methods to train the keypoint estimation networks in a self-supervised manner, e.g., Thewlis et al., 2019, the proposed methods incorporates the semantic consistency loss that makes the networks achieve a semantic awareness.
+ But, results on several benchmarks were clearly state-of-the-arts.

** Paper Weakness **

- Even though this paper was tailored to semi-supervision learning, unsupervised losses themself contribute the performance gains solely. So, I'm really interested in what happens the networks are trained only with unsupervised losses, i.e., SC and TC. It would be interesting because some datasets might not have any ground truth keypoints. In Table 2 and 3, the authors tried to provide an ablation study relevant to this, but the case of percentage of labeled images as 0 would be interesting.
- Conceptually, the transformation equivariance loss in (2) is similar to equivariant landmark transformation loss proposed by Honari et al. (2018) and recent other variants. The authors argue the proposed loss in (2) are different with them in that the previous methods leverage an inverse transformation. But, the methodological improvement is marginal and incremental.
- Transformation invariance loss in (4) is also interesting, but the loss itself might not contribute the performance gain. For example, if z and z' converge to 0, the loss is going to be minimum, but the networks are not trained well.
- In addition, because the paper solves the semi-supervised learning, the weights of loss components are important hyper-parameters, but there lack the ablation study for those.

---

> ### Author Response · Authors · 2020-11-20
> **Updated ablation study and additional sensitivity analysis**
>
> We thank the reviewer for the constructive feedback and suggestions. We have addressed all the reviewer’s concerns and conducted additional ablation studies and sensitivity analysis experiments. The paper is revised accordingly with all the requested results added.
>
> 1. As acknowledged in our submission, the transformation equivariance (TE) loss in Eq. (2) is inspired by the work of Honari et al., CVPR’18. Therefore, we did not claim this loss as our methodological contribution. However, we demonstrate that our modified TE loss can experimentally achieve better results compared to the loss of Honari et al., CVPR’18. We conducted additional experiments by replacing our TE loss with an inverse transformation loss of Honari et al. in our framework, and applied it on ATRW (tigers) and CUB-200-2011 (birds) datasets with 20% of labeled data. We observed that the score decreased by 1% on both datasets.
>
> 2. In the updated ablation study (Table 2 of the paper), we show experimentally that transformation invariance (4) contributes to the performance gain. As opposed to the original submission where the transformation equivariance and invariance are combined as TC (transformation consistency), in the updated ablation study, we treat them separately and evaluate the contribution of each constraint on the performance individually. The results of ablation study are added to Table 2.
>
> **Updated Table 2**
>
> | Unsupervised losses 	| 5% 	| 10% 	| 20% 	| 50% 	| 100% 	|
> |-	|:-:	|:-:	|:-:	|:-:	|:-:	|
> | TE + TI + SC 	| 66.32 	| 69.09 	| 71.62 	| 72.19 	| 74.44 	|
> | TE + TI 	| 46.76 	| 55.18 	| 64.01 	| 67.54 	| 72.11 	|
> | SC 	| 64.74 	| 67.43 	| 69.65 	| 70.61 	| 72.85 	|
> | TI + SC 	| 65.23 	| 68.11 	| 70.12 	| 71.28 	| 73.56 	|
> | TE + SC 	| 65.78 	| 68.51 	| 70.56 	| 71.77 	| 73.89 	|
> | TI 	| 43.62 	| 53.74 	| 61.12 	| 65.32 	| 71.80 	|
> | Supervised baseline 	| 39.16 	| 44.36 	| 54.23 	| 61.73 	| 71.91 	|
>
> 3. We conducted the proposed sensitivity analysis and evaluated the influence of the weights of loss components $(\lambda_1, \lambda_2, \lambda_3, \lambda_4)$ on the LSP dataset with 50% labeled data. We fixed the weight of supervised loss $\lambda_1 = 10^3$ and tested the following weights of unsupervised losses: $\lambda_2 =  (0.1, 0.5, 1.), \lambda_3 =  (10^1, 10^2, 10^3)$ and $\lambda_4 = (10^1, 10^2, 10^3)$. The ranges of weight values are different due to differences in scales for mean squared error of normalized pixels and cross-entropy loss. Experiments show that our method is not sensitive to variations of TE ($\lambda_3$) and TI ($\lambda_4$) losses (Figure 3 in the revised paper). The most notable drop in accuracy is observed when the weight of SC loss ($\lambda_2$) is reduced to 0.1 and the accuracy is at the same level when $\lambda_2$ equals 0.5 and 1.0.  We select the combination of $(\lambda_2, \lambda_3, \lambda_4) = (0.5, 10^2, 10^2)$ that achieves the highest score.
>
> We provide a table with the results of experiments because we cannot embed an image in comments. The paper is updated with Figure 3 which visually represents the results from the table:
>
> $\lambda_2$    |         $\lambda_3$       |   $\lambda_4$    |     Score|
> - | - | - | - |
> 0.1    |     10^1    |   10^1 |     71.55
> 0.1    |     10^1    |   10^2   |   71.78
> 0.1    |     10^1    |   10^3   |   72.05
> 0.1    |     10^2   |    10^1  |    71.56
> 0.1    |     10^2    |   10^2  |    71.63
> 0.1    |     10^2    |   10^3   |   71.76
> 0.1    |     10^3   |    10^1   |   71.48
> 0.1   |      10^3   |    10^2  |    71.57
> 0.1   |      10^3   |    10^3    |  71.88
> 0.5   |      10^1   |    10^1 |     72.14
> 0.5   |      10^1   |    10^2   |   72.25
> 0.5   |      10^1   |    10^3  |    72.52
> 0.5   |      10^2   |    10^1   |   72.47
> 0.5   |      10^2   |    10^2   |   72.63
> 0.5   |    10^2   |    10^3  |    72.56
> 0.5   |      10^3   |    10^1   |   72.45
> 0.5   |      10^3   |    10^2   |   72.49
> 0.5   |      10^3   |    10^3   |   72.26
> 1.0   |      10^1   |    10^1   |   72.44
> 1.0   |      10^1   |    10^2   |   72.45
> 1.0   |      10^1   |    10^3   |   72.41
> 1.0   |      10^2   |    10^1   |   72.41
> 1.0   |      10^2   |    10^2   |   72.34
> 1.0   |      10^2   |    10^3   |   72.31
> 1.0   |      10^3   |    10^1   |   72.40
> 1.0   |      10^3   |    10^2   |   72.44
> 1.0   |      10^3   |    10^3   |   72.42
>
> 4. We are not sure if the proposed study with only unsupervised losses can be directly applicable and comparable to the semi-supervised methods, and as such, to our approach. The evaluation in semi-supervised methods is based on the comparison against the ground truth, while in unsupervised methods, the detected keypoints are semantically different from the ones identified by human annotators.

---

### Decision · Program_Chairs · 2021-01-07
**Final Decision**

**Decision:**

Accept (Poster)

**Comment:**

This paper received three borderline reviews (2+ / 1-) and one positive review.  Having read through the reviews and author responses, the AC recommends the paper to be accepted.  The method, while simple, is proven experimentally to be effectively and will add to the body of work on key-point localization.   The authors are requested to add their additional baselines in the response text to the revision of their paper if it has not already been done.